# Mesenchymal Stem Cell-Derived Long Noncoding RNAs in Cardiac Injury and Repair

**DOI:** 10.3390/cells12182268

**Published:** 2023-09-13

**Authors:** Talan Tran, Claudia Cruz, Anthony Chan, Salma Awad, Johnson Rajasingh, Richard Deth, Narasimman Gurusamy

**Affiliations:** 1Department of Pharmaceutical Sciences, Barry and Judy Silverman College of Pharmacy, Nova Southeastern University, 3200 South University Drive, Fort Lauderdale, FL 33328, USA; 2Department of Bioscience Research, University of Tennessee Health Science Center, 847 Monroe Avenue, Memphis, TN 38163, USA

**Keywords:** mesenchymal stem cells, exosome, noncoding RNA, long noncoding RNA, heart failure, myocardial remodeling, cardiac regeneration

## Abstract

Cardiac injury, such as myocardial infarction and heart failure, remains a significant global health burden. The limited regenerative capacity of the adult heart poses a challenge for restoring its function after injury. Mesenchymal stem cells (MSCs) have emerged as promising candidates for cardiac regeneration due to their ability to differentiate into various cell types and secrete bioactive molecules. In recent years, attention has been given to noncoding RNAs derived from MSCs, particularly long noncoding RNAs (lncRNAs), and their potential role in cardiac injury and repair. LncRNAs are RNA molecules that do not encode proteins but play critical roles in gene regulation and cellular responses including cardiac repair and regeneration. This review focused on MSC-derived lncRNAs and their implications in cardiac regeneration, including their effects on cardiac function, myocardial remodeling, cardiomyocyte injury, and angiogenesis. Understanding the molecular mechanisms of MSC-derived lncRNAs in cardiac injury and repair may contribute to the development of novel therapeutic strategies for treating cardiovascular diseases. However, further research is needed to fully elucidate the potential of MSC-derived lncRNAs and address the challenges in this field.

## 1. Introduction

Cardiac injury, such as myocardial infarction and heart failure, remains a leading cause of mortality and morbidity worldwide [1]. The limited regenerative capacity of the adult mammalian heart poses a significant challenge for restoring its function after injury [2]. Therefore, understanding the underlying mechanisms of cardiac injury and repair is crucial for developing effective therapeutic strategies.

Cardiac injury typically occurs because of reduced blood supply, often due to the occlusion of a coronary artery, leading to ischemia and subsequent myocardial infarction [3]. Ischemic injury triggers a cascade of events, including inflammation, oxidative stress, cell death, and extracellular matrix remodeling, that contribute to tissue damage and functional impairment [2,4]. The heart possesses intrinsic mechanisms to initiate repair following injury. These processes involve the coordinated response of various cell types, signaling pathways, and extracellular matrix remodeling [5,6]. Upon injury, resident cardiac cells, such as cardiomyocytes, fibroblasts, and endothelial cells, undergo phenotypic changes and contribute to the repair process [5,7].

In recent years, mesenchymal stem cells (MSCs) have emerged as promising candidates for cardiac regeneration [8]. MSCs are multipotent stem cells with the capacity to differentiate into various cell types, including cardiomyocytes, endothelial cells, and smooth muscle cells. Additionally, MSCs exhibit immunomodulatory and paracrine effects, secreting a range of bioactive molecules that can modulate the cardiac microenvironment and promote tissue repair [9,10]. Recently, to further elucidate the mechanisms underlying the regenerative potential of MSCs in the context of cardiac injury and repair, attention has been given towards noncoding RNAs such as microRNAs (miRs) and long noncoding RNAs (lncRNAs). LncRNAs are a class of RNA molecules longer than 200 nucleotides that do not encode proteins but play critical roles in gene regulation and cellular processes. Emerging evidence suggests that MSC-derived lncRNAs may contribute to cardiac repair and regeneration by modulating key signaling pathways and cellular responses.

The role of MSC-derived miRNA in cardiac injury and repair has been widely studied [11,12]. However, there is limited knowledge on the role of lncRNAs in this field. In this review article, we aim to provide an overview of the current knowledge regarding the role of MSC-derived lncRNAs and their molecular mechanisms in cardiac injury and repair. By examining the characteristics of MSC-derived lncRNAs in cardiac injury models, we will shed light on their potential therapeutic applications and discuss the challenges and prospects in this exciting field.

## 2. Molecular Mechanisms of Cardiac Injury and Repair

### 2.1. Cardiac Injury

Cardiac injury triggers a cascade of cellular and molecular events that contribute to tissue damage and subsequent repair processes. Following an insult, such as myocardial infarction or cardiac ischemia, cardiomyocytes undergo necrosis or apoptosis, leading to the release of damage-associated molecular patterns (DAMPs). DAMPs, in turn, activate the innate immune response, leading to the recruitment of immune cells, including neutrophils, macrophages, and lymphocytes, to the injured area [2,13]. The infiltrating immune cells release proinflammatory cytokines, such as interleukin-1β (IL-1β) and tumor necrosis factor-alpha (TNF-α), exacerbating tissue damage and inflammation [6,13]. Concurrently, resident cardiac fibroblasts become activated, transitioning to a myofibroblast phenotype, and contribute to extracellular matrix remodeling by depositing collagen and other matrix components [14,15]. This remodeling process leads to scar formation, which initially provides structural support but can eventually impair cardiac function and contribute to adverse remodeling [2,16].

### 2.2. Cardiac Repair

Cardiac repair and regeneration involve intricate signaling pathways that regulate cellular processes crucial for angiogenesis, cardiomyocyte proliferation, fibroblast activation, and immune cell modulation. Key signaling pathways implicated in these processes include transforming growth factor-beta (TGF-β), Wnt/β-catenin, vascular endothelial growth factor (VEGF), and Notch signaling. TGF-β signaling regulates fibroblast activation, extracellular matrix synthesis, and myofibroblast differentiation, and its dysregulation can lead to pathological fibrosis and impaired cardiac function [17,18]. The Wnt/β-catenin pathway contributes to cardiac regeneration by promoting cardiomyocyte proliferation [19]. It plays a critical role in cardiac development and adult tissue homeostasis. VEGF signaling is pivotal in angiogenesis, facilitating the formation of new blood vessels to supply oxygen and nutrients to the injured myocardium [20]. It also participates in progenitor cell recruitment and activation. Notch signaling controls cell fate determination, proliferation, and differentiation during embryonic development and tissue repair [21]. Activation of Notch signaling promotes cardiomyocyte proliferation and angiogenesis in the injured heart [22]. 

## 3. Role of MSCs in Cardiac Regeneration and Repair

MSCs have garnered significant attention as potential therapeutic agents for cardiac regeneration and repair. MSCs are multipotent stem cells that can be isolated from various sources, including bone marrow, adipose tissue, and umbilical cord. They possess unique characteristics that contribute to their therapeutic potential, making them attractive candidates for cardiac tissue regeneration [23,24].

### 3.1. Paracrine Effects

One of the key mechanisms underlying the beneficial effects of MSCs in cardiac repair is their paracrine activity. MSCs secrete a range of bioactive molecules, including growth factors, cytokines, chemokines, and extracellular vesicles, which can modulate the cardiac microenvironment and promote tissue repair processes [25,26,27]. They secrete factors, such as vascular VEGF, hepatocyte growth factor, and insulin-like growth factor-1 (IGF1), and are shown to promote angiogenesis, reduce inflammation, inhibit apoptosis, and promote cell survival [23]. 

### 3.2. Immunomodulatory Properties

MSCs possess immunomodulatory properties that can modulate the immune response and create a favorable environment for cardiac repair. MSCs have been shown to suppress the activation and proliferation of immune cells, such as T cells, B cells, and natural killer cells, and inhibit the production of proinflammatory cytokines [9,28]. This immunomodulatory effect can attenuate excessive inflammation, which is detrimental to cardiac tissue, and it promotes a more balanced immune response conducive to tissue repair and regeneration [28].

### 3.3. Differentiation Potential

While the extent of MSC differentiation into functional cardiomyocytes in vivo is limited, MSCs have shown the capacity to differentiate into other cell types relevant to cardiac repair. MSCs can differentiate into endothelial cells, smooth muscle cells, and fibroblasts, contributing to neovascularization, tissue remodeling, and scar formation [23,29]. The paracrine factors secreted by MSCs also promote the differentiation and survival of endogenous cardiac progenitor cells, supporting the regeneration of functional myocardium [26].

### 3.4. Extracellular Matrix Remodeling

MSCs can also contribute to extracellular matrix remodeling, an essential process in cardiac repair. MSCs produce and secrete matrix metalloproteinases and tissue inhibitors of metalloproteinases, which regulate extracellular matrix turnover and remodeling [25]. This remodeling activity plays a critical role in scar formation, tissue healing, and overall cardiac tissue architecture.

Numerous preclinical studies and clinical trials have demonstrated the therapeutic potential of MSCs in cardiac regeneration and repair [30,31]. These studies have shown improvements in cardiac function, reduction in infarct size, increase in neovascularization, and attenuation of adverse remodeling [32]. However, challenges such as optimal cell delivery strategies, cell retention, and long-term survival of transplanted MSCs still need to be addressed to maximize their therapeutic efficacy.

## 4. MSC-Derived Exosomes and Noncoding RNAs

### 4.1. Exosomes

MSC-derived exosomes have emerged as a promising tool in regenerative medicine, particularly in the context of cardiac regeneration and repair [33,34]. Exosomes are small (30–150 nm in size) extracellular vesicles that play a crucial role in intercellular communication and the transport of various molecules within the body. These vesicles are released by cells and contain a diverse cargo of proteins, lipids, nucleic acids, and other bioactive molecules [35]. These exosomes possess trophic and immunomodulatory properties that mimic the effects of their parent cells, making them an attractive alternative to cell-based therapies [36]. Studies have demonstrated that MSC-derived exosomes can replicate the beneficial effects of MSCs, including anti-inflammatory, antiapoptotic, proangiogenic, and antifibrotic effects.

The generation of exosomes, which are extracellular vesicles characterized by a lipid bilayer membrane structure, begins with the initial budding of the cell membrane to form early endosomes, which are packaged with proteins and lipids known as multivesicle bodies (MVBs), that fuse with lysosomes to be degraded and recycled or fuse with the plasma membrane to be exported as exosomes [37,38]. For eukaryotic cells, this MVB pathway is indeed a major transmembrane protein and lipid turnover system that is extensively regulated by protein complexes involved in ubiquitination and the endosomal sorting complex required for transport (ESCRTs) [39]. After secretion, the exported exosomes can go on to deliver their diverse contents to nearby recipient cells through cytosol fusion, endocytosis, or ligand interactions [40]. 

While MSC therapy has shown promise in the treatment of cardiovascular disease, it has been associated with certain limitations, including immune rejection, tumorigenicity, and infusion toxicity [31]. However, MSC-derived exosomes have emerged as promising therapeutic agents due to their unique properties and ability to transfer various bioactive molecules, including noncoding RNAs such as miRs, lncRNAs, and circular RNAs [27]. Immunomodulatory and regenerative effects mediated through the secretion of biological nanoparticles within extracellular vesicles provide a unique therapeutic avenue to gain some of the benefits of MSCs [41]. 

### 4.2. miRNAs

Noncoding RNAs, such as miRNAs, lncRNAs, and circular RNAs, play crucial roles in regulating gene expression and cellular processes. In recent years, noncoding RNAs have gained greater interest due to studies implicating them in the regulation of MSC proliferation and differentiation [42]. miRNAs are a class of small noncoding RNAs that play a crucial role in the post-transcriptional regulation of gene expression. They are approximately 21–25 nucleotides in length and are involved in diverse biological processes, including development, differentiation, cell proliferation, apoptosis, and immune responses [43]. MiRNAs are transcribed from specific genes in the genome, forming primary miRNA transcripts (pri-miRNAs). These pri-miRNAs are processed by the Drosha enzyme in the nucleus, resulting in the production of precursor miRNAs (pre-miRNAs) [44]. Pre-miRNAs are then exported from the nucleus to the cytoplasm by Exportin-5. In the cytoplasm, the enzyme Dicer further processes pre-miRNAs into mature miRNAs [45]. Mature miRNAs are then incorporated into the RNA-induced silencing complex (RISC), where they guide the RISC to target messenger RNAs (mRNAs) through base pairing with complementary sequences in the mRNA molecules. This leads to translational repression or degradation of the target mRNAs, thereby regulating gene expression.

MSC-derived exosomes offer important favorable features as a noncellular alternative to MSCs themselves, especially since they avoid unwanted lineage differentiation [46]. Extracellular vesicle (EV) cargo can be modified to enrich specific miRNAs and growth factors, making them a good candidate for treatment of cardiac injury. This is achieved by upregulating the expression of specific angiogenesis-related miRNAs (miR-135b or miR-210) using lentiviral vectors carrying miR-135b or miR-210, thereby increasing the therapeutic properties of the MSCs [47]. Myocardial fibrosis, inflammation, apoptosis, and cardiac inflammation can therefore be ameliorated with the introduction of MSC-derived EVs carrying specific miRs, such as miR-302d-3p, which inhibit key inflammatory pathways [48,49,50]. Since the current review is focused on examining MSC-derived lncRNAs in cardiac injury and repair, for specific information regarding the role of MSC-derived exosomal miRNA, we recommend readers refer to the review of Nasser et al. [11]. 

Circular RNA is another emerging class of noncoding RNAs. Although not well studied, they are located in specific subcellular compartments and have been implicated in cardiac repair [51,52].

## 5. LncRNAs

LncRNAs have attracted considerable attention in recent years due to their diverse functions and regulatory roles in cellular processes [53,54]. LncRNAs are characterized by their extended length, typically exceeding 200 nucleotides. Unlike protein-coding mRNAs, lncRNAs lack an open reading frame capable of encoding functional proteins [55]. Instead, lncRNAs exhibit a wide range of structural features, including linear, circular, and complex secondary structures [56]. LncRNAs generally exhibit lower levels of evolutionary conservation compared to protein-coding genes, indicating their potential for rapid evolution and functional divergence [57]. Often, lncRNAs exhibit tissue-specific expressions that underscore their potential as biomarkers and therapeutic targets for various diseases, including cardiovascular disorders. 

### 5.1. LncRNA Biogenesis

LncRNAs undergo a complex process of biogenesis, involving transcription, post-transcriptional modifications, and maturation. The majority of lncRNAs are transcribed by RNA polymerase II from distinct genomic loci [58,59]. Similar to protein-coding genes, lncRNA loci are characterized by the presence of promoters, enhancers, and regulatory elements that control their transcription [60]. Following transcription, lncRNA transcripts undergo various post-transcriptional modifications that can influence their stability, processing, and subcellular localization [60]. These modifications include capping, splicing, polyadenylation, and RNA editing. Some lncRNAs can undergo alternative splicing, leading to the generation of multiple transcript isoforms with distinct functions [60,61].

### 5.2. Subcellular Localization and Functions of LncRNA

Many lncRNAs are processed and matured in the nucleus before their functional deployment. Nuclear processing steps include 5’ capping, splicing, and polyadenylation, which contribute to lncRNA stability, localization, and interactions with other molecules [62]. Nuclear-localized lncRNAs can participate in various nuclear processes, including transcriptional regulation, chromatin organization, and nuclear compartmentalization. Once processed in the nucleus, lncRNAs are exported to the cytoplasm through specific export machinery, such as the nuclear pore complex. In the cytoplasm, lncRNAs can exert their regulatory functions through interactions with RNA-binding proteins, miRNAs, and translation machinery components [63]. Cytoplasmic lncRNAs can influence mRNA stability, translation, and cellular signaling pathways. LncRNAs can be subjected to degradation pathways mediated by ribonucleases or undergo stabilization through binding to RNA-binding proteins or forming RNA-protein complexes [64]. 

LncRNAs exhibit remarkable functional diversity, participating in a wide range of biological processes and regulatory networks [62]. They can act as molecular scaffolds, guides, decoys, or enhancers, interacting with DNA, RNA, and proteins to modulate gene expression, chromatin organization, and cellular signaling pathways [62]. LncRNAs engage in diverse regulatory mechanisms, including cis- and trans-activation of genes. Cis-activation involves lncRNAs regulating nearby genes on the same chromosome, exemplified by XIST’s role in X-chromosome inactivation through chromatin modification and gene silencing [65]. Trans-activation, on the other hand, sees lncRNAs influencing gene expression on distinct chromosomes. For example, HOTAIR orchestrates gene silencing via histone modification by recruiting chromatin-modifying complexes [66], and MALAT1 influences gene networks by interacting with pre-mRNA processing factors and modulating alternative splicing mechanisms [67]. LncRNAs can also regulate miRs and the stability and translation of mRNAs, influence protein–protein interactions, and contribute to epigenetic regulation (Table 1).

## 6. The Role of MSC-Derived lncRNAs in Cardiac Injury and Repair

Recent studies reveal MSC-derived lncRNAs as promising candidates for cardiac regeneration and repair due to their ability to modulate the cardiac microenvironment. In this section, we discuss the importance of MSC-derived lncRNAs and their implications for cardiac regeneration and repair (Figure 1).

### 6.1. LncRNA-TARID Improves Cardiac Function

MSC-derived EVs have been shown to be beneficial in improving adverse myocardial remodeling and myocardial fibrosis in mouse models of myocardial infarction [68]. The LncRNA termed as TARID (Tcf21 antisense RNA inducing demethylation) is known to activate Tcf21 expression by inducing its promoter demethylation [85]. Tcf21, a basic helix-loop transcription factor for cardiovascular development, is a potential target for enhancing cardiac remodeling [86]. Zhu et al. [68] demonstrated that lncRNA-TARID-laden lipid nanoparticles increased Tcf21 expression, leading to improved cardiac function and histology in murine and porcine models of myocardial infarction.

### 6.2. LncRNA HAND2-AS1 Protects against Cardiomyocyte Injury

Hypoxia/reoxygenation (H/R) injury in H9c2 cells reduced the expression of lncRNA HAND2 antisense RNA 1 (HAND2-AS1) and induced the expression of miR-17-5p [69]. Previously, it was identified that the expression of miR-17-5p was elevated during myocardial injury and cardiomyocyte apoptosis [87]. Bone marrow-MSC-derived exosomes containing HAND2-AS1 improved cell viability, reduced apoptosis, controlled oxidative stress, and repressed inflammation, attenuating H/R-induced damage in H9c2 cells; however, HAND2-AS1 knockdown partially reversed the beneficial effects of exosomes, highlighting the significance of the HAND2-AS1/miR-17-5p axis in H/R-injured myocardial cells [69]. 

### 6.3. LncRNA A2M-AS1 Attenuates Myocardial Injury

LncRNA A2M antisense RNA 1 (A2M-AS1) was found to be lowly expressed in patients with acute myocardial infarction and in H/R-induced cardiomyocytes [71]. Exosomes derived from lncRNA A2M-AS1-transfected MSCs provided protective effects to H/R-induced cardiomyocytes [88]. The study suggested that exosomal delivery of lncRNA A2M-AS1 ameliorates H/R-induced cardiomyocyte apoptosis and oxidative stress by sponging miR-556-5p and increasing X-linked inhibitor of apoptosis protein (XIAP), providing insights into the pathogenesis of myocardial ischemia/reperfusion (I/R) injury [88].

### 6.4. Inhibition of LncRNA ZFAS1 Improves Myocardial Infarction

LncRNA zinc finger antisense 1 (ZFAS1) was shown to be an independent predictor of myocardial infarction [89] and to accelerate cardiomyocyte apoptosis in myocardial infarction in mice caused by calcium overload [90,91]. Xiao et al. [72] studied the role of lncRNA ZFAS1 in both in vitro and in vivo models of MI. Treatment with MSC-derived EVs improved cardiomyocyte viability; increased the expression of Von Willebrand factor and VEGF; and activated the Akt, Nrf2, and Heme oxygenase-1 pathways; however, overexpression of lncRNA ZFAS1 reversed these effects of MSC-derived EVs [72]. The studies suggest that the inhibition of ZFAS1 and activation of the Akt/Nrf2/Heme oxygenase-1 pathway by MSC-derived EVs could potentially improve myocardial infarction outcomes.

### 6.5. LncRNA Mir9-3hg Mitigates Cardiac Injury by Inhibiting Ferroptosis

Exosomes derived from bone marrow MSCs were shown to have high-level expression of the lncRNA MIR9-3 host gene (Mir9-3hg) and to promote cell proliferation while increasing glutathione levels and reducing reactive oxygen species and ferroptosis in cardiomyocytes subjected to H/R injury [73]. Additionally, it has been demonstrated that bone marrow-MSC-derived exosomes containing lncRNA Mir9-3hg improve cardiac function by modulating the RNA-binding protein Pum2 and peroxiredoxin 6, eventually inhibiting cardiomyocyte ferroptosis in mice with I/R injury [73].

### 6.6. LncRNA HCP5 in Mediating Cardioprotection

LncRNA HLA complex P5 (HCP5), present in the exosomes of human bone marrow MSCs, was identified as a key player in mediating protective effects such as enhanced cardiomyocyte viability and decreased apoptosis during H/R [76]. HCP5 is known to interact with miR-497, which targets IGF1 [76,92]. MSC-derived exosomes containing HCP5 sponged miR-497, leading to activation of the IGF1, PI3K, and AKT pathways and ultimately protected against cardiac I/R injury [76]. 

### 6.7. LncRNA UCA1 Protects against Cardiac Injury

Circulating lncRNA urothelial carcinoma-associated 1 (UCA1) may serve as a biomarker for acute myocardial infarction [93,94]. In addition, UCA1 was associated with electropathology in patients with atrial fibrillation [95]. Diao et al. [77] studied the role of lncRNA UCA1 present in exosomes derived from human umbilical cord MSCs and found that UCA1 protects against H/R injury in cardiac microvascular endothelial cells using a rat model of I/R injury. The UCA1 in exosomes competitively bound to miR-143, upregulated Bcl-2 expression, and led to the protection of cardiac microvascular endothelial cells against H/R injury [77].

### 6.8. LncRNA NEAT1 Protects against Cardiac Injuries

Several studies have shown that lncRNA nuclear paraspeckle assembly transcript 1 (NEAT1) plays an important role in cardiac diseases [96,97,98]. Exosomes derived from macrophage migration inhibitory factor (MIF)-pretreated MSCs (MIF-exosomes) were shown to protect mouse cardiomyocytes from hydrogen peroxide-mediated cell death [79]. This cardioprotective role of MIF-pretreated MSC exosomes was attributed to lncRNA NEAT1 via regulation of miR-142-3p and activating FOXO1 to promote cardiomyocyte survival and to inhibit apoptosis [79]. In another study, MIF-pretreated MSC exosomes recovered cardiac function and reduced cellular senescence through the transfer of NEAT1, which inhibits miR-221-3p, activates Sirt2, and counteracts the cardiotoxic effects of doxorubicin [80]. The study results demonstrate the cardioprotective roles of lncRNA NEAT1 found in MIF-pretreated MSC exosomes. 

### 6.9. LncRNA KLF3-AS1 Attenuates Myocardial Infarction

Exosomal lncRNA KLF3-AS1 derived from ischemic cardiomyocytes was shown to play a crucial role in mediating the secretion of IGF-1 by MSCs, thereby rescuing myocardial ischemia/reperfusion injury in both in vivo and in vitro models [99]. The lncRNA KLF3-AS1 in exosomes derived from human MSCs was shown to attenuate myocardial infarction by sponging miR-138-5p and activating Sirt1 expression leading to the inhibition of cellular apoptosis and pyroptosis [83].

### 6.10. LncRNA MALAT1 Prevents Aging-Induced Cardiac Dysfunction

LncRNA metastasis-associated lung adenocarcinoma transcript 1 (MALAT1) has been found to be associated with the modulation of myocardial ischemic injury, atherosclerotic progression, cellular senescence, and systemic inflammation [100,101,102]. It shows promise as a potential biomarker for acute myocardial infarction [103]. It was found that exosomes derived from human umbilical cord MSCs prevent aging-induced cardiac dysfunction [82]. Silencing lncRNA MALAT1 blocked the beneficial effects of these exosomes, suggesting its involvement in inhibiting the NF-κB/TNF-α signaling pathway [82]. The study results indicate that MSC-derived exosomes release lncRNA MALAT1, which prevents aging-induced cardiac dysfunction and provides insights into potential therapies for delaying aging and age-related diseases.

### 6.11. MSC-Pretreatment-Induced lncRNAs in Mediating Cardioprotection

Various pretreatment methods such as hypoxic preconditioning, treatment with either macrophage migration inhibitory factor (MIF) or atorvastatin, have been shown to facilitate the release of various lncRNAs from exosomes of MSCs and to promote cardiac repair by regulating various microRNAs (miR) and their signaling (Figure 2). 

Hypoxia preconditioning of human MSCs promoted the generation of exosomes containing lncRNA UCA1 that helped in improving cardiomyocyte survival and cardiac function in a rat model of myocardial infarction (MI) [78]. Mechanistically, it was found that UCA1 targeted miR-873 via sponging, leading to XIAP activation, AMPK phosphorylation, and increased expression of the antiapoptotic protein BCL2 [78]. In another study, lncRNA MALAT1 present in exosomes derived from hypoxia-preconditioned MSCs acted as a competing endogenous RNA that binds to miR-92a-3p, leading to ATG4a activation and improved mitochondrial metabolism, and ultimately improving doxorubicin-induced cardiomyopathy [81]. These findings suggest that exosomes derived from hypoxia-preconditioned MSCs could be a potential therapeutic option for cardiomyopathy.

Exosomes derived from atorvastatin-pretreated MSCs showed enhanced cardioprotective characteristics, promoting endothelial cell function, preventing cardiomyocyte apoptosis, and accelerating migration and tubelike structure formation [84]. The involvement of lncRNA H19 as a mediator of these effects was identified, regulating the expression of miR-675, proangiogenic factor VEGF, and intercellular adhesion molecule-1, indicating the enhanced therapeutic potential of atorvastatin-pretreated MSC-derived exosomes in treating acute myocardial infarction by promoting endothelial cell function [84].

### 6.12. LncRNA XIST Improves Atrial Fibrillation

Elevated levels of lncRNA X-inactive specific transcript (XIST) have been observed in patients with chronic heart failure, indicating its regulatory role in cardiomyocyte function [104]. Overexpression of lncRNA XIST in EVs derived from adipose tissue MSCs reduced myocardial pyroptosis and inflammation in atrial fibrillation mouse models and atrial myocytes [75]. XIST acted as a competing endogenous RNA (ceRNA) of miR-214-3p, promoting the upregulation of its target gene Arl2, an ADP-ribosylation factor [75]. These results suggest that lncRNA XIST can be a potential therapeutic target for atrial fibrillation.

### 6.13. LncRNA MIR155HG Improves Vascular Health

Sudden cardiac death risk has been linked to a functional indel polymorphism in the lncRNA MIR155 host gene (MIR155HG) [105]. Overexpression of lncRNA MIR155HG in MSCs enhanced their survival, migration, and antiapoptotic properties [70]. These protective effects extended to exosomes derived from MIR155HG-overexpressing MSCs, which improved the activity of human umbilical vein endothelial cells, mitigating intimal hyperplasia and ultimately protecting vascular endothelial integrity in a vein graft model using rats [70].

### 6.14. Inhibition of lncRNA LOC100129516 Promotes Cholesterol Efflux and Alleviates Atherosclerosis

LncRNA LOC100129516 levels were found to be upregulated in macrophage-derived foam cells induced by oxidized low-density lipoproteins [74]. In an ApoE-/- atherosclerosis mouse model, exosomal delivery of small interfering RNA against LOC100129516 decreased total cholesterol and low-density lipoprotein levels through activation of the peroxisome proliferator-activated receptor γ (PPARγ)/liver X receptor α (LXRα)/phospholipid-transporting ATPase ABCA1 (ABCA1) signaling pathway, promoting cholesterol efflux and suppressing intracellular lipid accumulation [74].

### 6.15. LncRNA Braveheart Promotes Cardiogenic Differentiation of MSCs In Vitro

lncRNA Braveheart interacts with Mesp1 in regulating the expression of cardiac transcription factors to promote cardiogenic differentiation [106,107]. Hou et al. [108] investigated the role of lncRNA Braveheart in promoting cardiogenic differentiation of MSCs in vitro. They found that transfection of lncRNA Braveheart into MSCs resulted in a higher percentage of differentiated cells with a cardiogenic phenotype compared to control groups through upregulation of cardiac-specific transcription factors (Nkx2.5, Gata4, Gata6, and Isl-1) and epithelial–mesenchymal transition-associated biomarkers (Mesp1, Snail, Twist, and N-cadherin) [108]. These findings suggest that lncRNA Braveheart promotes the trans-differentiation of MSCs into cardiogenic cells by enhancing the expression of cardiac-specific transcription factors and epithelial–mesenchymal transition-associated genes including Mesp1.

In addition to the above-mentioned MSC-derived lncRNAs, other lncRNAs have been identified as playing a critical role in transcriptional regulation and epigenetic control in cardiac development and cardiovascular diseases. LncRNA CARMN is known to trans-activate the myocardin/serum response factor complex, orchestrating smooth muscle cell differentiation and thwarting atherosclerotic neointima growth [109,110,111]. The lncRNA OIP5-AS1 was implicated in sex-specific differences in mitochondrial function and the development of heart failure, particularly exacerbating heart failure in female mice under cardiac pressure overload conditions [112]. LncRNAs like Trdn-as impact mRNA splicing and stability, as exemplified by their interaction with splicing factors to enhance efficient splicing of critical genes like Triadin, essential for calcium handling in cardiomyocytes [113]. Additionally, lncRNAs such as ZNF593-AS intercede in splicing processes, affecting excitation–contraction coupling in cardiomyopathy by modulating the splicing of the RYR2 gene [114].

The lncRNA myocardial infarction-associated transcript (MIAT) functions as a novel regulator in advanced atherosclerosis, controlling proliferation, apoptosis, and phenotypic transition of smooth muscle cells, along with proinflammatory properties of macrophages [115]. The lncRNA cardiac ischemia/reperfusion-associated Ku70 interacting lncRNA (CIRKIL) serves as a detrimental factor in myocardial I/R injury by regulating nuclear translocation of Ku70 and DNA double-strand breaks repair [116]. The novel vascular endothelial-associated lncRNA VEAL2 regulates endothelial permeability by competing with diacylglycerol for interaction with protein kinase C beta-b (Prkcbb) and modulating its kinase activity [117]. The cytoplasmic lncRNA Caren (short for cardiomyocyte-enriched noncoding transcript) maintains cardiac function by regulating translation of a distant gene, activating mitochondrial bioenergetics, and inactivating the ataxia telangiectasia mutated (ATM)-DNA damage response pathway to ensure cardiomyocyte homeostasis and cardioprotection under pathological stress [118].

### 6.16. The Role of MSC-Derived circRNA in Cardiac Injury and Repair

circRNAs have emerged as significant regulators in cardiac injury and repair, offering insights into potential therapeutic avenues for myocardial conditions. The cytoplasmic circRNA CircRTN4 was found to alleviate cardiac injury, apoptosis, and oxidative stress in a rat model of sepsis-induced myocardial injury. Further, it was observed that circRTN4 interacted with miR-497-5p to upregulate MG53 expression in cardiomyocytes, thereby mitigating cardiomyocyte damage [51]. In another study, circ-0001273 delivered by human umbilical cord MSC-derived exosomes inhibited myocardial cell apoptosis in ischemic conditions, promoting MI repair [52]. These studies collectively highlight the therapeutic potential of circRNAs, such as CircRTN4 and circ-0001273, delivered via exosomes, for preventing myocardial injury and promoting cardiac repair. Ruan et al. identified differentially expressed circRNAs that could potentially play roles during the differentiation of human umbilical cord MSCs into cardiomyocyte-like cells [119]. The circular RNA circHIPK3 plays a role in regulating cardiomyocyte senescence and cardiac function by acting as a scaffold for p21 mRNA-binding protein HuR and E3 ubiquitin ligase β-TrCP, promoting ubiquitination and degradation of HuR, reducing p21 activity, and influencing cellular senescence and cardiac dysfunction [120]. The comprehensive understanding of circRNAs’ regulatory functions in cardiac injury and repair provided by these studies contributes to advancing therapeutic strategies for myocardial conditions. 

## 7. Challenges and Future Prospects for Utilizing MSC-Derived lncRNAs in Clinical Practice

The therapeutic potential of MSC-derived lncRNAs in cardiac regeneration has opened up new avenues for innovative therapies. However, several challenges need to be addressed before the clinical translation of MSC-derived lncRNAs becomes a reality. 

To ensure consistent and reproducible outcomes, it is crucial to establish standardized protocols for MSC isolation, culture, and characterization. Variations in isolation techniques, culture conditions, and characterization methods can lead to heterogeneous MSC populations with varying regenerative capacities and lncRNA expression profiles [121]. The identification of key MSC-derived lncRNAs involved in cardiac regeneration is an ongoing challenge. Integrating high-throughput sequencing approaches, functional genomics, and bioinformatics analyses can aid in the discovery and characterization of MSC-derived lncRNAs with therapeutic potential. Additionally, understanding the underlying mechanisms through which these lncRNAs exert their effects will provide crucial insights for targeted interventions.

Efficient and targeted delivery of MSC-derived lncRNAs to the injured heart is critical for therapeutic success. Strategies that enhance tissue specificity, improve delivery efficiency, and minimize potential adverse effects need to be developed [121,122]. Before clinical translation, rigorous safety assessments are necessary to evaluate potential toxicities, off-target effects, and long-term consequences of MSC-derived lncRNA-based therapies. Furthermore, integration of MSC-derived lncRNAs with other regenerative strategies, such as gene therapy, cell-based therapies, and tissue engineering, may offer synergistic effects for enhanced cardiac regeneration.

The mobilization and activation of endogenous MSCs has been emphasized for tissue maintenance and repair. These cells are thought to participate in the body’s natural repair mechanisms in response to injury or disease. The presence of circulating endogenous MSCs has been reported in multiple pathophysiological conditions, and recent research has highlighted their potential as key components of interorgan communication networks [123]. However, further studies are required to understand the significance of such circulation endogenous MSCs and their secretomes containing lncRNAs.

Despite these challenges, the field of MSC-derived lncRNAs in cardiac regeneration holds immense potential. Future research efforts should focus on addressing the aforementioned challenges and advancing the field by elucidating the underlying mechanisms, conducting robust preclinical studies, optimizing delivery strategies, and establishing regulatory frameworks for clinical translation.

## 8. Conclusions

MSCs have emerged as promising candidates for cardiac regeneration due to their ability to differentiate into various cell types and secrete bioactive molecules. MSC-derived exosomes containing various miRs and lncRNAs have shown promise in regenerative medicine and can replicate the beneficial effects of MSCs by modulating various signaling pathways and cellular responses involved in cardiac repair. Recent studies have demonstrated the role of specific MSC-derived lncRNAs, such as TARID, HAND2-AS1, MIR155HG, A2M-AS1, and ZFAS1, in improving cardiac function and attenuating myocardial injury. These findings suggest that MSC-derived lncRNAs hold great promise for developing effective therapeutic strategies for cardiac regeneration and repair. However, further research is needed to fully understand the molecular mechanisms underlying the role of MSC-derived lncRNAs and to overcome the challenges associated with their therapeutic application.

## Figures and Tables

**Figure 1 cells-12-02268-f001:**
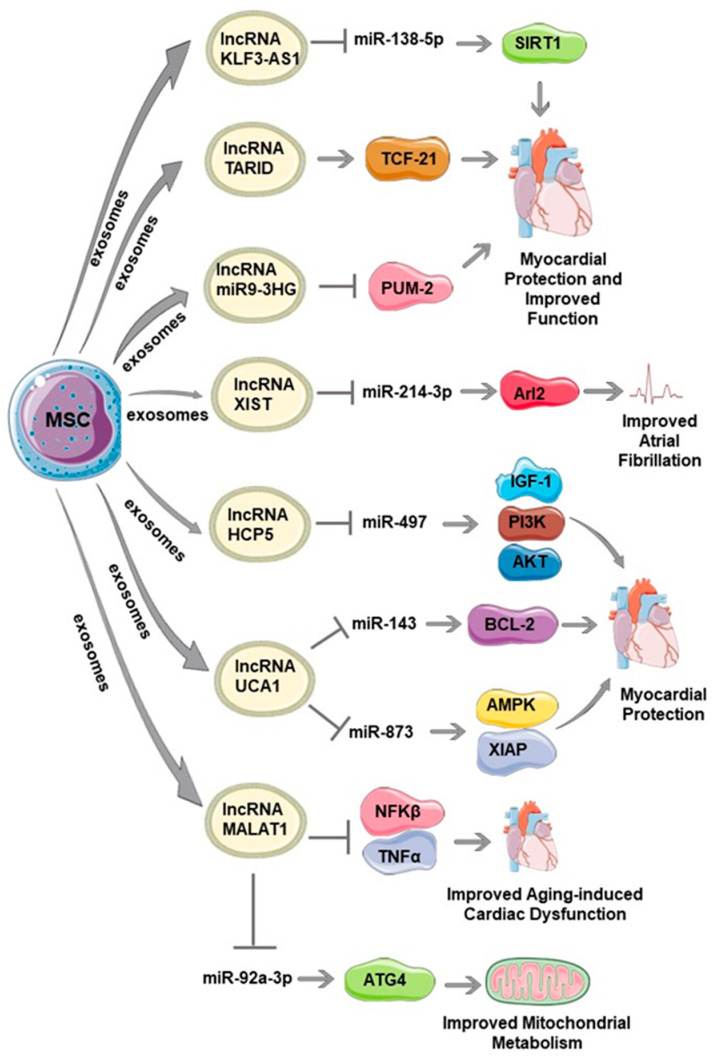
The role of mesenchymal stem cell (MSC)-derived long noncoding RNAs (lncRNAs) in cardiac repair. Several lncRNAs such as KLF3 AS-1, TARID, MIR9-3HG, XIST, HCP5, UCA1, and MALAT1 derived from the exosomes of MSCs mediate cardiac repair by regulating various microRNAs (miR) and signaling molecules such as NFκB, TNFα, AMPK, BCL2, etc. The figure was partly generated using Servier Medical Art, provided by Servier, licensed under a Creative Commons Attribution 3.0 unported license.

**Figure 2 cells-12-02268-f002:**
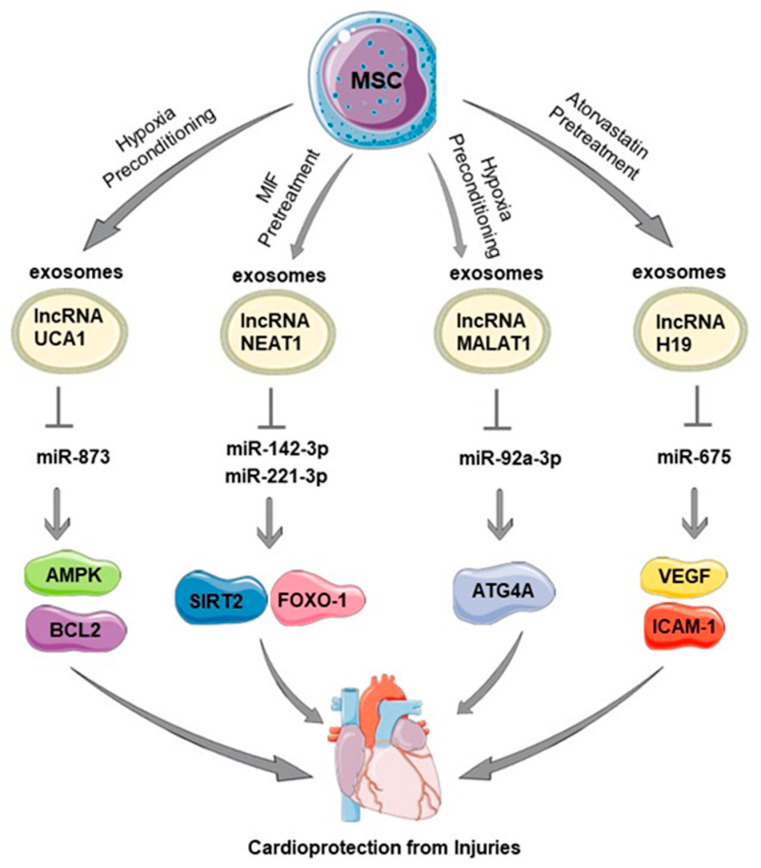
Pretreatment of mesenchymal stem cells (MSCs) mediates cardioprotection via long noncoding RNAs (lncRNA). Pretreatments such as hypoxic preconditioning, that is, treatment with either macrophage migration inhibitory factor (MIF) or atorvastatin, facilitate release of lncRNAs such as UCA1, NEAT1, MALAT1, and H19 in MSC exosomes and promote cardiac repair by regulating various microRNAs (miR) and signaling molecules such as AMPK, BCL2, SIRT2, FOXO, ATG4, VEGF, etc. The figure was partly generated using Servier Medical Art, provided by Servier, licensed under a Creative Commons Attribution 3.0 unported license.

**Table 1 cells-12-02268-t001:** The role of mesenchymal stem cell (MSC)-derived long noncoding RNAs (lncRNAs) in cardiac injury or repair.

Name of the Long Noncoding RNA (lncRNA)	Source of Mesenchymal Stem Cells (MSCs) and Exosomes	Experimental Model (In Vitro and or In Vivo)	Mechanism and the Role of LncRNA in Cardiac Injury or Repair	Reference
Tcf21 antisense RNA inducing demethylation (TARID)	MSC-derived extracellular vesicles (EVs)	In vivo: mouse and porcine models of myocardial infarction.	LncRNA-TARID to upregulate Tcf21 expression and improved cardiac function and myocardial fibrosis in mouse and porcine models of myocardial infarction.	[68]
HAND2 antisense RNA 1 (HAND2-AS1)	Bone marrow MSC-derived exosomes	In vitro: hypoxia/reoxygenation (H/R) injury in H9c2 cells.	LncRNA HAND2-AS1 protects against cardiomyocyte injury from H/R-induced apoptosis, oxidative stress, and inflammation by attenuating the expression of miR-17-5p.	[69]
MIR155 host gene (MIR155HG)	Exosomes derived from MIR155HG-overexpressing MSCs	In vivo: rat model of vein graft.	MSCs overexpressed with lncRNA MIR155HG protected vascular endothelial integrity, reduced inflammation, and significantly attenuated intimal hyperplasia.	[70]
A2M antisense RNA 1 (A2M-AS1)	Exosomes derived from lncRNA A2M-AS1-transfected MSCs	In vitro: hypoxia/reoxygenation (H/R) injury in human cardiomyocytes.	Exosomal delivery of lncRNA A2M-AS1 ameliorates H/R-induced cardiomyocyte apoptosis and oxidative stress through the regulation of the miR-556-5p/XIAP pathway.	[71]
Zinc finger antisense 1 (ZFAS1)	Exosomes from human bone marrow MSCs	In vitro: hypoxia injury in H9c2 cells. In vivo: rat models of myocardial infarction.	Treatment with MSC-derived EVs improved cardiomyocyte viability, increased expression of vWF and VEGF, and activated the Akt/Nrf2/HO-1 pathway. However, overexpressing lncRNA ZFAS1 reversed these effects of MSC-derived EVs.	[72]
MIR9-3 host gene (Mir9-3hg)	Exosomes derived from mouse bone marrow MSCs	In vitro: hypoxia/reoxygenation (H/R)-injury in HL-1 mouse cardiomyocytes. In vivo: mice ischemia/reperfusion injury.	Mir9-3hg was shown to bind with Pum2 protein and downregulate Pum2 expression. BM-MSC-derived exosomes ameliorated cardiac function in mice subjected to I/R injury by inhibiting cardiomyocyte ferroptosis through modulation of the Pum2/peroxiredoxin 6 (PRDX6) axis.	[73]
LOC100129516	Human bone marrow MSC-derived exosomes	In vitro: THP-1 cells were treated with oxidized low-density lipoproteins to induce foam cell formation. In vivo: ApoE−/− atherosclerosis mouse model.	Knockdown of MSC-derived exosomal lncRNA LOC100129516 promotes cholesterol efflux and alleviates atherosclerosis peroxisome proliferator-activated receptor γ (PPARγ)/liver X receptor α (LXRα)/phospholipid-transporting ATPase ABCA1 (ABCA1) signaling pathway.	[74]
X-inactive specific transcript (XIST)	EVs derived from mouse adipose tissue MSCs	In vitro: mouse HL-1 atrial myocytes. In vivo: mouse models of atrial fibrillation.	LncRNA XIST improves atrial fibrillation in vivo and in vitro. XIST acted as a competing endogenous RNA (ceRNA) of miR-214-3p, promoting the upregulation of its target gene Arl2.	[75]
HLA complex P5 (HCP5)	Exosomes from human bone marrow MSC cells	In vitro: hypoxia/reperfusion injury in human cardiac myocyte cell line. In vivo: rat model of myocardial I/R.	MSC-derived exosomes containing lncRNA HCP5 sponged miR-497, leading to activation of the IGF1/PI3K/AKT pathway and protection against I/R injury.	[76]
Urothelial carcinoma-associated 1 (UCA1)	Exosomes derived from human umbilical cord MSCs	In vitro: hypoxia/reoxygenation injury in cardiac microvascular endothelial cells. In vivo: rat model ischemia/reperfusion injury.	The lncRNA UCA1 competitively binds to miR-143, upregulating Bcl-2 expression and leading to the protection of cardiac microvascular endothelial cells against cardiac injury.	[77]
UCA1	Exosomes derived from hypoxia-conditioned human MSCs	In vivo: rat model of myocardial infarction.	UCA1 targets miR-873 via sponging, leading to XIAP activation, AMPK phosphorylation, and increased expression of the antiapoptotic protein BCL2.	[78]
Nuclear paraspeckle assembly transcript 1 (NEAT1)	Exosomes derived from macrophage migration inhibitory factor (MIF)-pretreated human-adipose-derived MSCs	In vitro: hydrogen peroxide treatment in human-iPSC-derived cardiomyocytes.	NEAT1 regulates the expression of miR-142-3p and activates FOXO1, thereby promoting cardiomyocyte survival and inhibiting apoptosis.	[79]
NEAT1	MIF-pretreated human-adipose-derived MSCs	In vivo: doxorubicin-induced cardiomyopathy.	MIF exosomes recover cardiac function and reduce cellular senescence through the transfer of NEAT1, which inhibits miR-221-3p, activates Sirt2, and counteracts the cardiotoxic effects of doxorubicin.	[80]
Metastasis-associated lung adenocarcinoma transcript 1 (MALAT1)	Exosomes derived from hypoxia-preconditioned human-adipose-derived MSCs	In vitro: doxorubicin treatment in human-iPSC-derived cardiomyocytes.	MALAT1 in exosomes derived from hypoxia-preconditioned MSCs acts as a ceRNA that binds to miR-92a-3p, leading to ATG4a activation and improved mitochondrial metabolism.	[81]
MALAT1	Exosomes derived from human umbilical cord MSCs	In vitro: hydrogen peroxide treatment in rat H9c2 cells. In vivo: mice model of aging.	MSC-derived exosomes release lncRNA MALAT1, which prevents aging-induced cardiac dysfunction by inhibiting the NF-κB/TNF-α signaling pathways.	[82]
KLF3 antisense RNA 1 (KLF3-AS1)	Exosomes from human MSCs	In vivo: rat model of myocardial infarction.	Overexpression of KLF3-AS1 in exosomes attenuates myocardial infarction by sponging miR-138-5p and regulates Sirt1 expression.	[83]
H19	Exosomes from atorvastatin-pretreated rat MSCs	In vivo: rat model of myocardial infarction.	LncRNA H19 acts as a mediator of the cardioprotective effects of atorvastatin-pretreated MSC-derived exosomes by regulating the expression of miR-675, proangiogenic factor VEGF, and intercellular adhesion molecule-1.	[84]

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
