# Peer review of "Mesenchymal Stem Cell-Derived Long Noncoding RNAs in Cardiac Injury and Repair"

_cells, 2023, doi:10.3390/cells12182268_

Round 1

Reviewer 1 Report

The authors have effectively summarized the role of lncRNAs in mesenchymal stem cell-mediated cardiac repair. Figures look great. Additionally, it might be beneficial for the authors to consider incorporating a section on circular RNAs, a subtype of long non-coding RNAs, to further enhance the comprehensiveness of their work.

Author Response

Dear Reviewer-1,

Comments 1: The authors have effectively summarized the role of lncRNAs in mesenchymal stem cell-mediated cardiac repair. Figures look great.

Response 1: The authors thank the Reviewer for the encouraging comments.

Comments 2: Additionally, it might be beneficial for the authors to consider incorporating a section on circular RNAs, a subtype of long non-coding RNAs, to further enhance the comprehensiveness of their work.

Response 2: We agree with this comment. Therefore, we have included the following in page 14.

6.16. The role of MSC-derived circRNA in cardiac injury and repair

circRNAs have emerged as significant regulators in cardiac injury and repair, offering insights into potential therapeutic avenues for myocardial conditions. A cytoplasmic circRNA CircRTN4 was found to alleviate cardiac injury, apoptosis, and oxidative stress in a rat model of sepsis-induced myocardial injury. Further, it was observed that circRTN4 interacted with miR-497-5p to upregulate MG53 expression in cardiomyocytes, thereby mitigating cardiomyocyte damage [51]. In another study, circ-0001273 delivered by human umbilical cord MSC-derived exosomes inhibited myocardial cell apoptosis in ischemic conditions, promoting MI repair [52]. These studies collectively highlight the therapeutic potential of circRNAs, such as CircRTN4 and circ-0001273, delivered via exosomes for preventing myocardial injury and promoting cardiac repair. Ruan et al identified differentially expressed circRNAs that could potentially play roles during the differentiation of human umbilical cord-MSCs into cardiomyocyte-like cells [116]. The circular RNA circHIPK3 plays a role in regulating cardiomyocyte senescence and cardiac function by acting as a scaffold for p21 mRNA-binding protein HuR and E3 ubiquitin ligase β-TrCP, promoting ubiquitination and degradation of HuR, reducing p21 activity, and influencing cellular senescence and cardiac dysfunction [117]. The comprehensive understanding of circRNAs' regulatory functions in cardiac injury and repair provided by these studies contributes to advancing therapeutic strategies for myocardial conditions.

Reviewer 2 Report

The authors have presented a well-summarized, comprehensive review of the roles of long non-coding RNAs (lncRNAs) derived from mesenchymal stem cells (MSCs) in cardiac injury and repair. They organized the article by opening with general molecular mechanisms of cardiac injury and repair followed by the role of MSCs in cardiac regeneration and repair, MSC-derived exosomes and non-coding RNAs, a focused background on lncRNAs, and importantly the role of various MSC-derived lncRNAs in cardiac injury and repair. Challenges and future prospects for utilizing MSC-derived lncRNAs in clinical practice were also discussed.

A brief discussion on the mobilization/activation of ‘endogenous’ MSCs and lncRNAs for cardiac repair and regeneration would be valuable (e.g., PMID: 33490065).

Since lncRNAs can act in cis or in trans, are any such differences known for the listed MSC-derived lncRNAs with respect to their roles in cardiac injury and repair? Including a note on this in the review would be relevant.

A brief statement on the comparison of MSC-derived lncRNAs versus lncRNAs derived from other sources in cardiac injury and repair would be valuable to provide a broader context.

Well written, however, minor editing would be helpful to clarify a few sentences.

Author Response

Dear Reviewer-2,

Comments 1: The authors have presented a well-summarized, comprehensive review of the roles of long non-coding RNAs (lncRNAs) derived from mesenchymal stem cells (MSCs) in cardiac injury and repair. They organized the article by opening with general molecular mechanisms of cardiac injury and repair followed by the role of MSCs in cardiac regeneration and repair, MSC-derived exosomes and non-coding RNAs, a focused background on lncRNAs, and importantly the role of various MSC-derived lncRNAs in cardiac injury and repair. Challenges and future prospects for utilizing MSC-derived lncRNAs in clinical practice were also discussed.

Response 1: The authors thank the Reviewer for the encouraging comments.

Comments 2: A brief discussion on the mobilization/activation of ‘endogenous’ MSCs and lncRNAs for cardiac repair and regeneration would be valuable (e.g., PMID: 33490065).

Response 2: Thank you for pointing out this interesting article. We agree with this. Therefore, we have included a paragraph in the revised manuscript in the section 7 (highlighted in Red) mentioning the significance of the mobilization of endogenous MSCs, as mentioned below.

“The mobilization and activation of endogenous MSCs has been emphasized for tissue maintenance and repair. These cells are thought to participate in the body's natural repair mechanisms in response to injury or disease. The presence of circulating endogenous MSCs has been reported in multiple pathophysiological conditions, and recent research has highlighted their potential as key components of interorgan communication networks [123]. However, further studies are required to understand the significance of such circulation endogenous MSC and their secretomes containing lncRNAs.”

Comments 3: Since lncRNAs can act in cis or in trans, are any such differences known for the listed MSC-derived lncRNAs with respect to their roles in cardiac injury and repair? Including a note on this in the review would be relevant.

Response 3: We agree with this comment. Unfortunately, the role of majority of the lncRNAs are emerging, and the specific Cis or Trans activation for the gene expression is not available for the majority of the lncRNAs. However, we have included this helpful point in the general introduction about the lncRNAs in section 5.2, as mentioned below.

“LncRNAs engage in diverse regulatory mechanisms, including cis- and trans-activation of genes. Cis-activation involves lncRNAs regulating nearby genes on the same chromosome, exemplified by XIST's role in X-chromosome inactivation through chromatin modification and gene silencing [65]. Trans-activation, on the other hand, sees lncRNAs influencing gene expression on distinct chromosomes. For example, HOTAIR orchestrates gene silencing via histone modification by recruiting chromatin-modifying complexes [66], and MALAT1 influences gene networks by interacting with pre-mRNA processing factors and modulating alternative splicing mechanisms [67].”

Comments 4: A brief statement on the comparison of MSC-derived lncRNAs versus lncRNAs derived from other sources in cardiac injury and repair would be valuable to provide a broader context.

Response 4: We agree with this comment. Therefore, we have included the following two paragraphs in page 13 and 14.

In addition to the above mentioned MSC-derived lncRNAs, other lncRNAs that have been identified to play a critical role in transcriptional regulation and epigenetic control in cardiac development and cardiovascular diseases. LncRNA CARMN was known to transactivate the myocardin/serum response factor complex, orchestrating smooth muscle cell differentiation and thwarting atherosclerotic neointima growth [106-108]. The lncRNA OIP5-AS1 was implicated in sex-specific differences in mitochondrial function and the development of heart failure, particularly exacerbating heart failure in female mice under cardiac pressure overload conditions [109]. LncRNAs like Trdn-as impact mRNA splicing and stability, as exemplified by their interaction with splicing factors to enhance efficient splicing of critical genes like Triadin, essential for calcium handling in cardiomyocytes [110]. Additionally, lncRNAs such as ZNF593-AS intercede in splicing processes, affecting excitation-contraction coupling in cardiomyopathy by modulating the splicing of the RYR2 gene [111].

The lncRNA myocardial infarction-associated transcript (MIAT) functions as a novel regulator in advanced atherosclerosis, controlling proliferation, apoptosis, and phenotypic transition of smooth muscle cells, along with proinflammatory properties of macrophages [112]. The lncRNA cardiac ischemia reperfusion associated Ku70 interacting lncRNA (CIRKIL) serves as a detrimental factor in myocardial I/R injury by regulating nuclear translocation of Ku70 and DNA double-strand breaks repair [113]. The novel vascular endothelial-associated lncRNA VEAL2 regulates endothelial permeability by competing with diacylglycerol for interaction with protein kinase C beta-b (Prkcbb) and modulating its kinase activity [114]. The cytoplasmic lncRNA Caren (short for cardiomyocyte-enriched noncoding transcript) maintains cardiac function by regulating translation of a distant gene, activating mitochondrial bioenergetics, and inactivating the ataxia telangiectasia mutated (ATM)-DNA damage response pathway to ensure cardiomyocyte homeostasis and cardioprotection under pathological stress [115].

Comments on the Quality of English Language

Well written, however, minor editing would be helpful to clarify a few sentences.

We have edited the manuscript for minor English language corrections.

Reviewer 3 Report

The manuscript titled "Mesenchymal stem cell-derived long non-coding RNAs in cardiac injury and repair" is a comprehensive review that explores the potential role of MSC-derived lncRNAs in cardiac regeneration. It effectively presents background information on cardiac injury, heart regeneration, and the contributions of MSCs to this process. The review covers various beneficial properties of MSCs, including their paracrine effects and immunomodulatory properties.

The section on the role of MSC-derived lncRNAs in cardiac injury and repair is well-structured, providing an in-depth overview of recent studies. It discusses a diverse range of lncRNAs implicated in different aspects of cardiac regeneration, such as improving cardiac function, protecting against cardiomyocyte injury, and attenuating myocardial infarction. Mechanistic insights, including miRNA sponging and pathway activation, enhance understanding of lncRNA functions.

Overall, the manuscript is well-researched and valuable, catering to regenerative medicine, cardiovascular biology, and molecular biology researchers. It sheds light on the emerging field of MSC-derived lncRNAs and their potential implications for cardiac repair.

Author Response

Dear Reviewer-3,

Comments: The manuscript titled "Mesenchymal stem cell-derived long non-coding RNAs in cardiac injury and repair" is a comprehensive review that explores the potential role of MSC-derived lncRNAs in cardiac regeneration. It effectively presents background information on cardiac injury, heart regeneration, and the contributions of MSCs to this process. The review covers various beneficial properties of MSCs, including their paracrine effects and immunomodulatory properties.

The section on the role of MSC-derived lncRNAs in cardiac injury and repair is well-structured, providing an in-depth overview of recent studies. It discusses a diverse range of lncRNAs implicated in different aspects of cardiac regeneration, such as improving cardiac function, protecting against cardiomyocyte injury, and attenuating myocardial infarction. Mechanistic insights, including miRNA sponging and pathway activation, enhance understanding of lncRNA functions.

Overall, the manuscript is well-researched and valuable, catering to regenerative medicine, cardiovascular biology, and molecular biology researchers. It sheds light on the emerging field of MSC-derived lncRNAs and their potential implications for cardiac repair.

Response: The authors thank the Reviewer for providing an excellent summary of our manuscript and for the encouraging comments.

Reviewer 4 Report

This article does not discuss the role of MSC-derived lncRNAs in cardiac injury. It is suggested to change the title to: 'Mesenchymal Stem Cell-Derived Long Non-Coding RNAs in Cardiac Regenesis and Repair'. Additionally, it is recommended to summarize part of the role of MSCs in cardiac regeneration and repair using icons. Furthermore, the article should consider adding a section on the different damage repair potential and application prospects of MSCs from various sources. Lastly, it is suggested to further classify the types of myocardial injury, such as myocardial ischemia, injury, and necrosis.

Minor editing of English language required

Author Response

Dear Reviewer-4,

Comments 1: This article does not discuss the role of MSC-derived lncRNAs in cardiac injury. It is suggested to change the title to: 'Mesenchymal Stem Cell-Derived Long Non-Coding RNAs in Cardiac Regenesis and Repair'.

Response 1: Although majority of the lncRNAs included in the manuscript involved in cardiac repair and regeneration, there are a few lncRNAs such as ZFAS1 (mentioned in section 6.4) and LOC100129516 (mentioned in section 6.14) play a role in inducing injuries. Thus, we believe that it is appropriate to keep the current tile “Mesenchymal Stem Cell-Derived Long Non-Coding RNAs in Cardiac Injury and Repair.”

Comments 2: Additionally, it is recommended to summarize part of the role of MSCs in cardiac regeneration and repair using icons.

Response 2: We thank the Reviewer for indicating this. Since the focus of the article was on MSC-derived lncRNAs, we have included diagrams only for MSC-derived lncRNAs. Considering the length and focus of this manuscript, we are unable to include a diagram summarizing the role of MSCs in cardiac regeneration and repair. However, the readers can refer to the diagrams regarding the role of MSCs in cardiac regeneration available in the referenced articles.  

Comments 3: Furthermore, the article should consider adding a section on the different damage repair potential and application prospects of MSCs from various sources.

Response 3: We have tried to include all the sources of MSC-derived lncRNAs including the MSC-derived exosomes. We have not restricted ourselves to any particular source of MSCs. We have included the following two paragraphs in page 13 and 14 mentioning the roles of various other lncRNAs (some of them are present in MSC, but did not mention here) that play important role in cardiac repair.

In addition to the above mentioned MSC-derived lncRNAs, other lncRNAs that have been identified to play a critical role in transcriptional regulation and epigenetic control in cardiac development and cardiovascular diseases. LncRNA CARMN was known to transactivate the myocardin/serum response factor complex, orchestrating smooth muscle cell differentiation and thwarting atherosclerotic neointima growth [106-108]. The lncRNA OIP5-AS1 was implicated in sex-specific differences in mitochondrial function and the development of heart failure, particularly exacerbating heart failure in female mice under cardiac pressure overload conditions [109]. LncRNAs like Trdn-as impact mRNA splicing and stability, as exemplified by their interaction with splicing factors to enhance efficient splicing of critical genes like Triadin, essential for calcium handling in cardiomyocytes [110]. Additionally, lncRNAs such as ZNF593-AS intercede in splicing processes, affecting excitation-contraction coupling in cardiomyopathy by modulating the splicing of the RYR2 gene [111].

The lncRNA myocardial infarction-associated transcript (MIAT) functions as a novel regulator in advanced atherosclerosis, controlling proliferation, apoptosis, and phenotypic transition of smooth muscle cells, along with proinflammatory properties of macrophages [112]. The lncRNA cardiac ischemia reperfusion associated Ku70 interacting lncRNA (CIRKIL) serves as a detrimental factor in myocardial I/R injury by regulating nuclear translocation of Ku70 and DNA double-strand breaks repair [113]. The novel vascular endothelial-associated lncRNA VEAL2 regulates endothelial permeability by competing with diacylglycerol for interaction with protein kinase C beta-b (Prkcbb) and modulating its kinase activity [114]. The cytoplasmic lncRNA Caren (short for cardiomyocyte-enriched noncoding transcript) maintains cardiac function by regulating translation of a distant gene, activating mitochondrial bioenergetics, and inactivating the ataxia telangiectasia mutated (ATM)-DNA damage response pathway to ensure cardiomyocyte homeostasis and cardioprotection under pathological stress [115].

Comments 4: Lastly, it is suggested to further classify the types of myocardial injury, such as myocardial ischemia, injury, and necrosis.

Response 4: We apologize to the Reviewer that although we have mentioned the specific type of injury such as myocardial ischemia, infarction, apoptosis, necrosis or fibrosis in specific places, we have not attempted to classify them. Also, for clarity we have commonly mentioned in the subtitles as cardiac injury for all the types of injuries.

Comments on the Quality of English Language

Minor editing of English language required.

We have edited the manuscript for minor English language corrections.